# 2D-TPE: Two-Dimensional Positional Encoding Enhances Table Understanding for Large Language Models

## Abstract

Tables are ubiquitous across various domains for concisely representing structured information. Empowering large language models (LLMs) to reason over tabular data represents an actively explored direction. However, since typical LLMs only support one-dimensional (1D) inputs, existing methods often flatten the two-dimensional (2D) table structure into a sequence of tokens, which can severely disrupt the spatial relationships and result in an inevitable loss of vital contextual information. In this paper, we first empirically demonstrate the detrimental impact of such flattening operations on the performance of LLMs in capturing the spatial information of tables through two elaborate proxy tasks. Subsequently, we introduce a simple yet effective positional encoding method, termed "2D-TPE" (Two-Dimensional Table Positional Encoding), to address this challenge. 2D-TPE enables each attention head to dynamically select a permutation order of tokens within the context for attending to them, where each permutation represents a distinct traversal mode for the table, such as column-wise or row-wise traversal. 2D-TPE effectively mitigates the risk of losing essential spatial information while preserving computational efficiency, thus better preserving the table structure. Extensive experiments across five benchmarks demonstrate that 2D-TPE outperforms strong baselines, underscoring the importance of preserving the table structure for accurate table comprehension. Comprehensive analysis further reveals the substantially better scalability of 2D-TPE to large tables than baselines. [1]

## CCS Concepts

• **Information systems** → **Information retrieval**; • **Computing methodologies** → **Artificial intelligence**.

## Keywords

table understanding, large language model, positional encoding

**ACM Reference Format:**
Anonymous Author(s). 2018. 2D-TPE: Two-Dimensional Positional Encoding Enhances Table Understanding for Large Language Models. In *Proceedings of Make sure to enter the correct conference title from your rights confirmation emai (Conference acronym 'XX)*. ACM, New York, NY, USA, 14 pages. https://doi.org/XXXXXXX.XXXXXXX

---

[1]Code and data are available at https://anonymous.4open.science/r/2D-TPE-797.

---

## 1 Introduction

Tables are highly structured and rich in information, making them indispensable and widely used in the real world. From financial reports to scientific data, tables serve as an efficient means of organizing and presenting complex relationships and patterns. As large language models (LLMs) continue to advance [1, 3, 46], and the interest in developing LLM-based agents for completing specific tasks grows [8, 19, 30, 45, 59], endowing LLMs with the ability to accurately comprehend and reason over tabular data has emerged as a crucial research direction [56, 58, 61].

A fundamental challenge for LLM-based table understanding lies in the inherent mismatch between the two-dimensional (2D) structure of tables and the one-dimensional (1D) input format required by LLMs. To bridge the gap, existing methods typically flatten the tabular data into a sequence of tokens [61]. While this simple approach offers a straightforward way to adapt tabular data to existing LLMs, it disregards the spatial relationships and contextual information encoded within the layout of tables. Consequently, LLMs may struggle to perform accurate analysis and reasoning over tabular data, even for seemingly simple tasks. For example, we devise two proxy tasks (as illustrated in Figure 1), namely *Counting-Stars* and *Locating-Values*, to assess the capability of LLMs to identify specific cells based on their positional relations to another cell, which is a crucial foundation for table understanding (more details of the tasks are presented in §3.3). Empirical evaluations demonstrate that LLMs equipped with conventional 1D positional encodings perform remarkably poorly on these tasks, achieving an accuracy of less than 5% and 20% in $20 \times 20$ tables, respectively. The results highlight the detrimental impact of flattening operations on preserving table structures and underscore the need for more effective encoding methods to facilitate LLMs' perception of tabular data.

In this work, we present a novel positional encoding approach, dubbed "2D-TPE" (Two-Dimensional Table Positional Encoding), designed to effectively capture both semantic and spatial information inherent in 2D tabular data while seamlessly accommodating 1D textual data. Akin to tables, images also convey information through points distributed across a 2D space, rendering spatial information critically important [31]. However, recent advancements in vision-language models (VLMs) employing 2D positional encoding [49] are not readily transferable to the table understanding domain. This is because tables exhibit a dynamic nature with varying sizes and variable-length tokens within individual cells, in contrast to the fixed-size patches used in image representations. Consequently, we argue that tables should be treated as a unique modality, distinct from both textual and image data, to fully leverage their inherent structure and spatial relationships. In a nutshell, 2D-TPE enables each attention head to dynamically select a permutation order for perceiving the context, where each permutation represents a distinct traversal mode for the table, such as column-wise or row-wise traversal. Through dynamic permutation selection, our approach

allows for flexible and adaptive context perception, enabling the model to explore various traversal patterns and capture the most relevant spatial dependencies. This adaptability is particularly valuable in scenarios where the importance of specific dimensions or relationships within the table may vary, ensuring that the model can effectively focus on the most salient aspects of the data. In this way, 2D-TPE can mitigate the risk of losing essential spatial information while maintaining computational efficiency.

Specifically, 2D-TPE employs an architecture where each attention head mixes up the attention outputs calculated using different permutation orders through a routing network that dynamically determines the routing weights, thereby capturing diverse perspectives of the spatial relationships between cells. We fine-tune the model by combining the standard language modeling loss with an auxiliary entropy minimization term, encouraging the model to distinctly leverage specific permutation orders for different attention heads and tokens. In this paper, we demonstrate the effectiveness of 2D-TPE using row-wise and column-wise traversal modes. Nevertheless, the proposed framework is flexible and can readily accommodate more permutation orders, such as diagonal traversal. This extensibility allows for systematic exploration of different inductive biases and spatial encoding strategies, potentially unlocking further performance gains in various table understanding tasks.

We conduct experiments with an open-source LLM on five benchmarks, covering a wide range of table understanding tasks, including question-answering, type annotation, relation extraction, and entity linking. The evaluation results consistently demonstrate the superiority of 2D-TPE over strong baselines employing the same LLM, with most improvements (3 out of 5 tasks) exhibiting statistical significance (Sign-test, $p$-value < 0.05). Notably, 2D-TPE exhibits exceptional robustness and scalability when confronted with tables of varying sizes, maintaining stable performance even when the table quadruples. These findings underscore the substantial potential of 2D-TPE in tackling real-world challenges involving large-scale tabular data. In stark contrast, the performance of 1D positional encoding deteriorates significantly as table sizes increase, highlighting their fragility in handling complex table structures. Remarkably, 2D-TPE achieves an excellent balance between efficacy and efficiency. Compared to vanilla Transformers, the additional computational cost in terms of TFLOPs and memory usage is negligible, with an increase of less than 2%. Furthermore, the inference time per example only experiences a modest 13% increase. In summary, 2D-TPE paves the way for more effective and versatile table understanding systems.

We summarize our contributions as follows:
**I.** We propose two proxy tasks to empirically demonstrate the detrimental impact of flattening 2D table structures into 1D sequences, highlighting the loss of vital spatial information.
**II.** We introduce a versatile positional encoding method "2D-TPE," which enables LLMs to dynamically select different permutation orders for perceiving the table's context, efficiently and effectively preserving the spatial relationships within the tabular data.
**III.** Through comprehensive experiments on five tabular tasks, we show that our proposed 2D-TPE method outperforms strong baselines. Further analysis illustrates a larger margin between 2D-TPE and baselines for larger tables, revealing its better scalability.

## 2 Related Works

### 2.1 LLM-based Table Understanding

Numerous researchers have endeavored to harness the remarkable capabilities of LLMs to tackle table understanding problems, including table question answering [37, 64], table augmentation [13], fact verification [2, 6], table interpretation [13] and table-to-text generation [36], by converting tables into 1D sequences of tokens.

*2.1.1 Instruction-Tuning for Table Understanding.* To tailor LLMs for table-related tasks, several studies have curated specialized tabular datasets for instruction-tuning purposes [26, 29]. For instance, Table-GPT [27] synthesized diverse instruction-completion pairs from real tables. TableLlama [58] introduced the comprehensive TableInstruct dataset, supporting varied tasks and showcasing a model with broad generalization across benchmarks.

*2.1.2 Prompt Engineering for Table Understanding.* LLMs have shown a remarkable reasoning capacity [52] through the Chain of Thought (CoT) prompting strategy [5, 53] that solves complex queries step by step. Consequently, considerable research efforts have been dedicated to developing various prompting techniques to improve LLM performance in table understanding tasks. For example, Dater [56] prompted the LLM to extract key sub-tables and decompose questions into sub-questions. TaCo [63] used the CoT approach in tabular LMs for mathematical queries. PROTRIX [54] introduced a Plan-then-Reason framework for structured problem-solving and step-by-step reasoning.

*2.1.3 Tool Usage for Table Understanding.* To address the structured nature and inherent logic of tabular data, several studies have explored the integration of LLMs with auxiliary tools such as SQL and Python interpreters, enabling precise calculations and location-based operations. The text-to-SQL paradigm [34] translated natural language queries into SQL for data retrieval and manipulation. Binder [10] integrated Python tools for complex computations and precise cell positioning in tables. ReAcTable [62] and Chain-of-Table [51] interleaved reasoning and tool invocation, enabling LLMs to dynamically use tools throughout the problem-solving process.

Orthogonal to the above studies, 2D-TPE takes a fundamentally different approach by addressing the intrinsic challenge of preserving the 2D table structure when encoding tabular data into the 1D input format required by LLMs.

### 2.2 Table Modeling

In addition to the efforts that directly transform tables into sequences as inputs for LLMs, some work focuses on designing model architectures to better handle the 2D structure of tabular data.

HyTrel [4] transformed tabular data into hypergraphs to capture structural attributes but lacked compatibility with mainstream LLMs. Recent studies have adapted attention mechanisms to accommodate the inherently 2D structure of tabular data. TABERT [57] layered column-wise self-attention on top of row-wise self-attention, enhancing positional awareness among tokens. StruBERT [47] employed a combination of horizontal and vertical self-attention. TURL [13] and MATE [17] restricted attention to tokens within the same row or column. TABLEFORMER [55] introduced learnable biases to adjust attention scores based on token positions.

Unlike these models, 2D-TPE efficiently encodes spatial relationships within the standard self-attention framework, better aligning with mainstream LLMs.

## 2.3 Positional Encodings

The attention mechanism in the vanilla Transformer [48] lacks the ability to capture inter-token positional relationships. To overcome this limitation, researchers have proposed absolute and relative positional encodings to incorporate positional information.

### 2.3.1 Absolute Positional Encoding (APE).

*1D APE.* One intuitive approach is to map position indices into learnable embeddings, as employed in the BERT [14] and GPT [41]. However, this method fails to generalize to positions that have not been encountered during training, leading to substantial performance degradation when the inference length exceeds the training length [43]. To address the challenge, Vaswani et al. [48] introduced the sinusoidal position embeddings that mapped a position index $m$ to a fixed embedding $P_m$ through a series of sinusoidal functions. Under this formulation, for any position offset $\Delta_m$, the positional embedding $P_{m+\Delta_m}$ for the token $m + \Delta m$ can be represented as a linear function of $P_m$, thereby facilitating the model's potentials to generalize patterns based on relative positions.

*2D APE.* Prior research also attempted to encode 2D tabular data using learnable embeddings. Among the efforts, TAPAS [21] used multiple positional embeddings per table token to denote row and column indices. TABBIE [24] combined outputs from two Transformers with unique positional embeddings to encode row and column contexts. TUTA [50] introduced tree-based embeddings for hierarchical table positions. Unlike 2D-TPE, the above methods with learnable positional embeddings still potentially face the challenges of length extrapolation.

### 2.3.2 Relative Positional Encoding (RPE).

*1D RPE.* RPE focuses on inter-token relative distances, enhancing the model's length extrapolation capability. The most commonly employed RPE techniques are ALiBi [39] and RoPE[43], both of which are applied on every self-attention layer without additional trainable parameters. ALiBi introduced a linear bias to each attention term. RoPE modulated the query and key vectors using rotary matrices derived from absolute position indices, with the attention weights remaining solely contingent on the relative positional offset between the query and key.

*2D RPE.* 2D RPE has been adapted for image encoding due to images' inherent 2D structure, requiring positional encodings for patch sequences. Unified-IO-2 [31] adapted RoPE to 2D by dividing the query and key vectors of attention heads and applying separate rotary embeddings from horizontal and vertical coordinates. Although effective for image tasks [20, 32], its application to tables is limited by inability to distinguish tokens within the same cell and independent dimensional attention processing, which may miss inter-token positional patterns. This issue is less critical in fixed-patch image encoding. In contrast, 2D-TPE uses varied permutation orders of tokens to capture structural table information, overcoming these issues and scaling to incorporate more permutation orders to capture more structural information beyond horizontal and vertical directions. While 2D-TPE in this study builds on RoPE, it can adapt to other RPE techniques (e.g., ALiBi).

## 3 Background: 1D Positional Encoding

In this section, we introduce the background of 2D-TPE, including the representative 1D positional encoding approach RoPE (§3.1), which has been widely adopted in state-of-the-art LLMs such as MiniCPM [23], Llama [46], etc; the limitation of 1D positional encoding for representing table structures (§3.2); and an empirical investigation to demonstrate the limitation (§3.3).

## 3.1 Rotary Position Embedding

Let us consider a Transformer model with $H$ attention heads, each with a dimension of $d$. Given a sequence $X = (x_1, x_2, \cdots, x_M)$ as input, the query vector of the $h$-th head for the token $x_m$ in a certain layer is represented as $q_m^h \in \mathbb{R}^d$, while the key and value vectors of the same head for the token $x_n$ are $k_n^h \in \mathbb{R}^d$ and $v_n^h \in \mathbb{R}^d$, respectively. The output $o_m$ of the self-attention module for $x_m$ is computed as the concatenation of the outputs from $H$ heads:

$$o_m = o_m^1 \oplus o_m^2 \oplus \cdots \oplus o_m^H, \tag{1}$$

where $\oplus$ denotes the vector concatenation operation, $o_m^h$ is a weighted sum of the values of the $h$-th head, where the weight assigned to each value is computed by a compatibility function $f$ between the query and the corresponding key:

$$o_m^h = \sum_{n \leqslant m} a_{m,n}^h v_n^h, \tag{2}$$

$$a_{m,n}^h = \frac{\exp\big(f(q_m^h, k_n^h)\big)}{\sum_{j \leqslant m} \exp\big(f(q_m^h, k_j^h)\big)}. \tag{3}$$

The core principle of RoPE is to integrate positional information into the query and key in the compatibility function $f$:

$$f(q_m^h, k_n^h) = (\hat{q}_m^h)^\top \hat{k}_n^h = (R_m^{b,d} q_m^h)^\top (R_n^{b,d} k_n^h) = (q_m^h)^\top R_{n-m}^{b,d} k_n^h, \tag{4}$$

where $R_m^{b,d}$ is a rotary matrix:

$$R_m^{b,d} = \begin{bmatrix} r_{m,1}^{b,d} & O & \cdots & O \\ O & r_{m,2}^{b,d} & \cdots & O \\ O & O & \cdots & r_{m,\frac{d}{2}}^{b,d} \end{bmatrix} \in \mathbb{R}^{d \times d}, \tag{5}$$

$$r_{m,i}^{b,d} = \begin{bmatrix} \cos m\theta_i^{b,d} & -\sin m\theta_i^{b,d} \\ \sin m\theta_i^{b,d} & \cos m\theta_i^{b,d} \end{bmatrix}, (i = 1, 2, \cdots, \frac{d}{2}) \tag{6}$$

where $\theta_i^{b,d} = b^{-\frac{2(i-1)}{d}}$ and $b$ is a fixed base angle. It is noteworthy that the compatibility score $f(q_m^h, k_n^h)$ only depends on the relative distance between the query and the key (i.e, $n - m$).

## 3.2 Limitation for Encoding Table Structures

1D positional encoding techniques have demonstrated their efficacy in various natural language processing tasks. However, when confronted with the intricate structure of tables, these methods exhibit a significant limitation—the loss of crucial spatial information.

**Figure 1: Illustration for the proposed two proxy tasks.**

This deficiency can potentially result in suboptimal representations, thereby complicating many fundamental table understanding tasks.

Specifically, when flattening a table into a 1D sequence, regardless of the traversal method used, the original spatial proximity of the table is compromised. For example, when using row-wise traversal, the relative distance between a cell and its vertically adjacent cells increases from an immediate proximity of 1 to the number of columns, and vice versa for column-wise traversal. Consequently, the model is burdened with the task of counting to determine whether cells are in the same row or column, potentially leading to a substantial loss of spatial information, particularly for large tables with greater distances between related cells.

### 3.3 Struggling with Table Understanding

To quantitatively demonstrate the limitation imposed by 1D positional encoding techniques, we devise two proxy tasks: *Counting-Stars* and *Locating-Values*. We aim to gain deeper insights into the weaknesses of 1D positional encoding for table understanding, thereby motivating the development of more robust and effective solutions that can leverage the spatial information in tables.

*3.3.1 Design Principles.* We craft the two tasks to assess the capability of LLMs to identify row and column information, which is fundamental for table understanding [44]. As illustrated in Figure 1, *Counting-Stars* evaluates the parallel lookup capability from the perspectives of both rows and columns, while *Locating-Values* targets the serial lookup capability, which demands multi-hop reasoning to locate an intermediate cell based on relative positional offsets.

*3.3.2 Task Description.* We describe the tasks in detail as follows:

*Counting-Stars.* Given a table and a reference number, the model must identify all cells containing a designated star symbol that are in the same row or column as the reference. This task requires thorough understanding of positional relationships across both dimensions of the table. We assess performance using the accuracy of the output list, with the order of elements being inconsequential. As exemplified in Figure 1 (a), we first fill the tables with stars, using numbers randomly selected from $1 \sim 9$ (with repetition allowed) and appended with a star symbol. Each row and column must contain $1 \sim 3$ starred cells. For the remaining cells, we populate them

**Table 1: Accuracy (%) on the proposed *Counting-Stars* and *Locating-Values* tasks with different table sizes, where $n \times n$ means the table have $n$ rows and $n$ columns.**

| Tasks | Methods | 10×10 | 15×15 | 20×20 |
|---|---|---|---|---|
| Counting-Stars | Row-wise Traversal | 90.46 | 6.05 | 0.05 |
| | Column-wise Traversal | 86.45 | 12.50 | 2.20 |
| | Constrained Attention | 0.00 | 0.00 | 0.00 |
| | **2D-TPE** | **99.65** | **98.70** | **89.75** |
| Locating-Values | Row-wise Traversal | 87.55 | 33.15 | 18.79 |
| | Column-wise Traversal | 90.25 | 37.25 | 15.70 |
| | Constrained Attention | 82.80 | 21.75 | 0.55 |
| | **2D-TPE** | **92.50** | **61.50** | **54.70** |

with unique integers uniformly sampled from the range $0 \sim 999$, ensuring that non-starred integers are unique, with one randomly selected as the reference number.

*Locating-Values.* Given a table and a lookup instruction, the model should output the target value from the table by following the provided instruction. We format the instruction as "What is the value $c$ columns to the right/left of and $r$ rows below/above ★?" ($r \neq 0, c \neq 0$). This formulation necessitates two-hop reasoning: identifying the correct row and then the specific column (or vice versa) to locate the target cell. We use accuracy for evaluation. Figure 1 (b) shows an example. Tables are populated with unique integers in $0 \sim 999$, with one cell designated by a star. The values of $r$, $c$, and the locating directions are randomly assigned. And the target cell always falls within the table's boundaries.

*3.3.3 Evaluated Methods.* To comprehensively evaluate the efficacy of 1D positional encodings for table understanding, we investigate three distinct methods: **(1) Row-wise Traversal:** It encodes token positions within a table by traversing sequentially across rows, assigning incremental positional encodings to each token encountered; **(2) Column-wise Traversal:** It employs a column-wise traversal strategy; And **(3) Constrained Attention:** It permits each token in the table to attend only to tokens residing within the same row or column based on row-wise traversal, while tokens in the text are able to attend to all others [13, 17]. We implement all the above methods based on MiniCPM [23] with causal self-attention and 1D RoPE. Furthermore, we also report the performance of the proposed 2D-TPE for reference. Details of 2D-TPE are left until §4.

*3.3.4 Experimental Setup.* We design three settings that encompass tables with varying sizes, specifically $10 \times 10$, $15 \times 15$, and $20 \times 20$, to investigate the scalability of the methods in handling tables of different sizes. For each setting, we automatically construct the training/validation/test data with 10,000/2,000/2,000 examples.

*3.3.5 Results and Insights.* As presented in Table 1, the results clearly demonstrate: **(1) 1D positional encoding methods achieve significantly lower accuracy than 2D-TPE across both tasks; and (2) the performance gap between 2D-TPE and 1D methods widens as the table size increases.** 1D methods even exhibit a near-complete loss of ability to accurately locate cells in $20 \times 20$ tables for *Counting-Stars*. This finding underscores the severe impact of losing 2D spatial information when using 1D positional encodings. While the constrained attention method attempts to

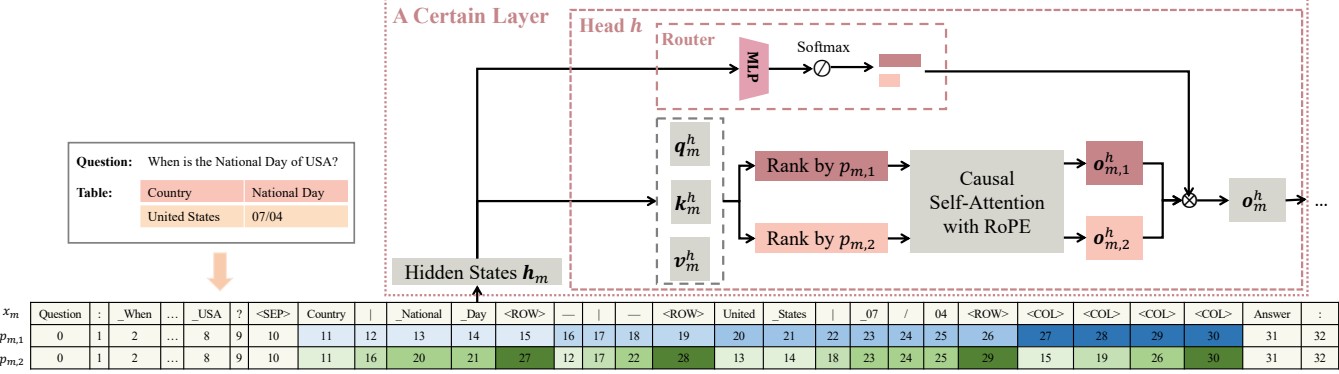

| $x_m$ | Question | : | _When | ... | _USA | ? | <SEP> | Country | \| | _National | _Day | <ROW> | — | \| | — | <ROW> | United | _States | \| | _07 | / | 04 | <ROW> | <COL> | <COL> | <COL> | <COL> | Answer | : |
|---|---|---|---|---|---|---|---|---|---|---|---|---|---|---|---|---|---|---|---|---|---|---|---|---|---|---|---|---|---|
| $p_{m,1}$ | 0 | 1 | 2 | ... | 8 | 9 | 10 | 11 | 12 | 13 | 14 | 15 | 16 | 17 | 18 | 19 | 20 | 21 | 22 | 23 | 24 | 25 | 26 | 27 | 28 | 29 | 30 | 31 | 32 |
| $p_{m,2}$ | 0 | 1 | 2 | ... | 8 | 9 | 10 | 11 | 16 | 20 | 21 | 27 | 12 | 17 | 22 | 28 | 13 | 14 | 18 | 23 | 24 | 25 | 29 | 15 | 19 | 26 | 30 | 31 | 32 |

**"<SEP>":** Separator between Table and Question     **"—" / "|":** Separator between Rows/Columns     **"<ROW>"/"<COL>":** Row/Column Ending

**Figure 2: Overview of 2D-TPE. $x_m$: the $m$-th token in the sequence; $p_{m,1}/p_{m,2}$: the position index for the token $x_m$ using row/column-wise traversal. The indices in the same color mean that their corresponding tokens are in the same row/column when using $p_{m,1}/p_{m,2}$, respectively.**

explicitly define rows and columns for each cell, its poor performance suggests a significant mismatch between this approach and the inherent attention mechanism employed during the pretraining of LLMs. The 2D-TPE method, on the other hand, leverages the spatial structure of tables in a more natural and effective manner.

In summary, the elaborate tasks serve as a rigorous testbed for evaluating the table understanding abilities, shedding light on the limitations of 1D positional encoding methods in capturing the inherent spatial information within tables. Their poor performance motivates us to design 2D positional encoding methods.

## 4 Methodology

Inspired by the previous analysis, we propose 2D-TPE by extending RoPE to encode 2D positional information. In this way, we can leverage its existing strengths while enabling LLMs to better perceive and reason about tabular data structures.

Formally, we define table understanding tasks as follows: Given a question $Q$ and a table $T$, the model should generate an answer $A$ to the question $Q$ by comprehending the information presented in $T$. To address the problem, we are inspired by the Mixture-of-Expert approach [25] to allow each attention head to perceive contextual information from various perspectives by dynamically selecting a permutation order over the table. Furthermore, we define a training objective (§4.2) to optimize the model. Our approach incorporates a carefully curated set of candidate permutation orders (§4.3), facilitating efficient exploration of the 2D table structure. Figure 2 provides an overview of the 2D-TPE framework.

### 4.1 Model Architecture

The model can perceive 2D information through multiple permutation orders over the table. To this end, we first concatenate the question $Q$, the table $T$, and a text-form instruction "Answer:" into a sequence of $M$ tokens, denoted as $X = (x_1, x_2, \cdots, x_M)^2$. Subsequently, we define the positional encodings for $X$ as $P = (\boldsymbol{p}_1, \boldsymbol{p}_2, \cdots, \boldsymbol{p}_M)$, where $\boldsymbol{p}_m = (p_{m,1}, p_{m,2}, \cdots, p_{m,J})$ is a vector,

---

$^2$The order of tokens in $X$ are inessential since we specify the position of each token $x_m$ explicitly in $P$ as $\boldsymbol{p}_m$.

and $p_{m,j}$ corresponds to the position index of $x_m$ in the $j$-th permutation order.

Taking $X$ into the model, we calculate the attention output $\boldsymbol{o}_m^h$ of the $h$-th head for $x_m$ in a certain self-attention layer as a mixture of attention outputs derived using different permutation orders:

$$\boldsymbol{o}_m^h = \sum_{j=1}^{J} r_{m,j}^h \boldsymbol{o}_{m,j}^h, \tag{7}$$

where $r_{m,j}^h$ and $\boldsymbol{o}_{m,j}^h$ are the routing weight and attention output corresponding to the $j$-th permutation order, respectively. We introduce an additional routing network to each self-attention layer to calculate the routing weights:

$$r_{m,j}^h = \mathrm{Softmax}(\mathrm{MLP}(\boldsymbol{h}_m^h))\big|_j, \tag{8}$$

where the MLP network projects the hidden state $\boldsymbol{h}_m^h \in \mathbb{R}^d$ of the $h$-th head for the token $x_m$ into logits over $J$ permutation orders$^3$:

$$\mathrm{MLP}(\boldsymbol{h}_m^h) = \boldsymbol{W}_{\mathrm{down}}(\mathrm{SiLU}(\boldsymbol{W}_{\mathrm{up}}\boldsymbol{h}_m^h) \odot (\boldsymbol{W}_{\mathrm{gate}}\boldsymbol{h}_m^h)), \tag{9}$$

where $\boldsymbol{W}_{\mathrm{up}} \in \mathbb{R}^{4d \times d}$, $\boldsymbol{W}_{\mathrm{gate}} \in \mathbb{R}^{4d \times d}$ and $\boldsymbol{W}_{\mathrm{down}} \in \mathbb{R}^{J \times 4d}$ are trainable weights, and SiLU is the activation function [18]. The design of the MLP network is aligned with the Llama models [46].

The attention output of the $j$-th permutation order is calculated using the standard causal self-attention network with 1D RoPE:

$$\boldsymbol{o}_{m,j}^h = \sum_{p_{n,j} \leqslant p_{m,j}} a_{m,n,j}^h \boldsymbol{v}_n^h, \tag{10}$$

$$a_{m,n,j}^h = \frac{\exp\big((\boldsymbol{q}_m^h)^\top \boldsymbol{R}_{p_{n,j}-p_{m,j}}^{b,d} \boldsymbol{k}_n^h\big)}{\sum_{p_{i,j} \leqslant p_{m,j}} \exp\big((\boldsymbol{q}_m^h)^\top \boldsymbol{R}_{p_{i,j}-p_{m,j}}^{b,d} \boldsymbol{k}_i^h\big)}, \tag{11}$$

where $a_{m,n,j}^h$ denotes the attention score between $x_m$ and $x_n$ in the $j$-th permutation order. As the attentive fields of different attention heads vary depending on the traversal modes, we re-rank the query, key, and value by the ascending order of token positions within each mode before applying the causal self-attention module. By

---

$^3$We obtain $\boldsymbol{h}_m^h$ by splitting the hidden state $\boldsymbol{h}_m$ into $h$ heads.

this means, each head can efficiently attend to the appropriate contextual information.

## 4.2 Training Objective

The standard language modeling loss aims to minimize the negative log-likelihood of ground-truth answers as follows:

$$\mathcal{L}_{\text{NLL}} = -\log P(A|Q, T). \tag{12}$$

Furthermore, in order to encourage the model to select a specific permutation order for each attention head and each token more distinctly, we introduce an auxiliary loss to sharpen the distribution of router weights by minimizing its entropy [7]:

$$\mathcal{L}_{\text{ENT}} = \frac{1}{MH} \sum_{m=1}^{M} \sum_{h=1}^{H} E_m^h, \tag{13}$$

$$E_m^h = -\sum_{j=1}^{J} r_{m,j}^h \log r_{m,j}^h. \tag{14}$$

In this way, the model can utilize information from one permutation order without interference from blending all. In summary, we train the model using the following objective:

$$\mathcal{L} = \mathcal{L}_{\text{NLL}} + \lambda \mathcal{L}_{\text{ENT}}, \tag{15}$$

where $\lambda$ is a tunable hyper-parameter.

## 4.3 Candidate Permutation Orders

Using proper permutation orders as candidates in 2D-TPE is a crucial consideration. Let us first investigate tokens within the table. One can traverse a table following different orders, such as row-wise, column-wise, diagonal, Hilbert-curve [22], and Z-order-curve [16] traversals, each of which induces distinct position indices and representing varying inductive biases regarding the proximity of cells within the table. In this paper, we illustrate the effect of 2D-TPE using two representative traversal modes to obtain the permutation orders (i.e., $J = 2$ in Equation 7): row-wise and column-wise traversals, both proceeding from top-left to bottom-right. This choice can be readily extended to accommodate other traversal modes, which is left for future work. Note that the relative distances between tokens in the same cell (e.g., "United" and "_States" in Figure 2) always remain the same regardless of permutation orders.

For tokens in the text interleaved with tables, we maintain their position indices consistent with the incremental position index along the text sequence in all permutation orders. During the generation process, we also incrementally assign position indices to generated tokens. Such design ensures that the attention mechanism between tokens within the text remains equivalent to the standard 1D RoPE, maintaining consistency with mainstream LLMs.

Through this systematic exploration of permutation orders, we aim to provide a principled framework for applying 2D-TPE to various scenarios involving both text and tabular data.

## 5 Experiments

## 5.1 Experimental Setup

*5.1.1 Evaluation Benchmarks.* To rigorously evaluate the performance of 2D-TPE, we conduct experiments on five diverse table understanding tasks, as summarized in Table 2. We curate all datasets from TableInstruct [58] and maintain only those examples with correct table structures and lengths not exceeding 4,096 tokens counted using the MiniCPM tokenizer [23].

As for the evaluation metrics, we adopt the official evaluation scripts for all datasets. Specifically, we use accuracy (**ACC** for short) for evaluation on HiTab and the entity-linking subset in TURL; use accuracy, recall [38], and Micro F1 for the relation extraction and column type annotation subsets in TURL. For FeTaQA, which has free-form answers, we use BLEU-4 [35] and ROUGE [28].

*5.1.2 Baselines.* We compare 2D-TPE against several aforementioned strong baselines using conventional 1D RoPE, including *Row-wise Traversal*, *Column-wise Traversal*, and *Constrained Attention*. Additionally, we compare 2D-TPE with TABBIE [24], which integrates row and column embeddings for enhanced table representations, and Multimodal Rotary Position Embedding (M-RoPE), originally used in Qwen2-VL [49] to jointly capture image and text positional information. We adapt M-RoPE to tabular data by decomposing the RoPE positional embeddings into three independent components for row, column, and cell positions.

*5.1.3 Implementation Details.* We train TABBIE on each dataset according to the settings described in [24] and implement 2D-TPE and other baselines by fine-tuning MiniCPM-2B-SFT [23] on each dataset, selected for its impressive performance [60] and ease of industrial deployment. We set the hyper-parameter $\lambda$ in Eq. 15 to 1, the batch size to 64, the learning rate to 2e-5, the length limits to 4,096 and the warm-up steps to 3% of 2 epochs. Appendix B further describes the influence of hyper-parameter settings in detail. We employ DeepSpeed with ZeRO-2 [42] and Flash-attention-2 [11] for all methods, except *Constrained Attention* that modifies the attention mask, and thus becomes incompatible with Flash-Attention. We use greedy decoding for inference.

## 5.2 Results

As shown in Table 3, **2D-TPE is superior to baselines across five datasets, particularly on HiTab, where tables are significantly larger than others.** The results reveal the importance of effectively leveraging spatial information. Notably, different datasets may require information in distinct dimensions. For example, *Row-wise Traversal* significantly outperforms other baselines on HiTab. Conversely, *Column-wise Traversal* excels in RelExtra. However, these baselines are inherently limited by their 1D perception of the table. Although M-RoPE achieved moderate performance on EntLink, FeTaQA, and ColType, it exhibited the poorest results on the HiTab dataset. This discrepancy may be because answers can solely be derived from the corresponding tables on HiTab, unlike other datasets that may rely on supplementary information from the questions. Consequently, M-RoPE's ability to comprehend table structures based on questions may be limited. In contrast, 2D-TPE allows for token-wise selection of the more valuable spatial dimension.

We notice that the superiority of 2D-TPE over baselines on RelExtra is less pronounced than on other tasks. Manual inspection reveals that many questions for this task are sufficiently informative to induce answers without extensive reasoning over the tables. On the other hand, the performance gains observed on these benchmarks are less substantial than those witnessed in our proposed

**Table 2: Statistics of evaluation benchmarks.**

| Task | Dataset | Training | Validation | Test | Avg. # Row | Avg. # Column | Avg. Table Length | Avg. Question Length |
|---|---|---|---|---|---|---|---|---|
| Hierarchical Table QA | HiTab [9] | 7K | 0.5K | 1K | 21.9 | 8.5 | 1,294 | 23 |
| Column Type Annotation | | 20K | 1K | 2K | 13.3 | 5.7 | 546 | 1,814 |
| Relation Extraction | TURL [13] | 54K | 1K | 1K | 19.3 | 5.5 | 829 | 2,307 |
| Entity Linking | | 20K | 1K | 1K | 21.0 | 4.9 | 803 | 996 |
| Highlighted Cells QA | FeTaQA [33] | 7K | 0.5K | 2K | 14.7 | 6.1 | 719 | 100 |

**Table 3: Experiment results of different methods. ↑ means the larger scores indicate a better performance. We highlight the best result in bold and underline the second best. * indicates that 2D-TPE significantly outperforms the baseline ($p < 0.05$ with Sign Test). "EntLink", "RelExtra," and "ColType" refer to the entity linking, relation extraction, and column type annotation subsets in the TURL dataset, respectively.**

| Method | HiTab (%) ↑ | EntLink (%) ↑ | FeTaQA (%) ↑ | | | | RelExtra (%) ↑ | | | ColType (%) ↑ | | |
|---|---|---|---|---|---|---|---|---|---|---|---|---|
| | ACC | ACC | BLEU-4 | ROUGE-1 | ROUGE-2 | ROUGE-L | ACC | Recall | F1 | ACC | Recall | F1 |
| Row-wise Traversal | 66.31* | 82.58* | 64.51 | 62.22 | 39.57 | 53.42 | 96.56 | 90.85 | 93.62 | 86.53* | 82.48 | 84.46* |
| Column-wise Traversal | 60.52* | 82.91* | 64.18 | 61.85* | 38.74* | 53.08* | 96.67 | 91.28 | 93.90 | 87.65* | 82.78 | 85.15 |
| Constrained Attention | 22.66* | 82.22* | 64.76* | 62.34 | 39.41 | 53.46 | 84.53* | 79.42* | 81.90* | 83.29* | 81.14* | 82.20* |
| TABBIE | 62.21* | 73.78* | 63.17* | 61.37* | 38.50* | 52.74* | 95.15* | 89.66* | 92.32* | 88.07* | 80.24* | 83.98* |
| M-RoPE | 14.39* | 83.33* | 64.62 | 62.46 | 39.62 | 53.65 | 95.75* | 90.09* | 92.83* | 88.83* | 81.77* | 85.16 |
| 2D-TPE | 68.19 | 84.10 | 65.70 | 63.54 | 40.59 | 54.71 | 96.83 | 91.66 | 94.18 | 89.79 | 83.77 | 86.68 |

**Table 4: Statistics and results for size scaling on HiTab, where $n + n$ means inserting $n$ table(s) to the left and right of the original one, respectively. Val is short for validation.**

| Setting | 0+0 | 1+1 | 2+2 |
|---|---|---|---|
| **Data Statistics** | | | |
| # Train/Val/Test | 3K/0.3K/0.6K | 3K/0.3K/0.6K | 3K/0.3K/0.6K |
| Avg. # Row | 11.9 | 11.9 | 11.9 |
| Avg. # Column | 8.1 | 21.9 | 35.8 |
| Avg. # Table Length | 657 | 1,647 | 2,637 |
| Avg. # Question Length | 23 | 23 | 23 |
| **Accuracy (%)** | | | |
| Row-wise Traversal | 57.65 | 34.40 | 27.52 |
| Column-wise Traversal | 30.12 | 25.38 | 24.46 |
| Constrained Attention | 11.01 | 7.95 | 6.73 |
| 2D-TPE | 59.48 | 56.73 | 54.13 |

proxy tasks in §3.3. This discrepancy may be attributed to the distinctiveness of rows and columns in these benchmarks: individual rows or columns possess unique identifiers or highly distinguishing features (e.g., "Date" vs. "Name"). This inherent distinctiveness facilitates easier cell location, potentially diminishing the advantages gained from preserving spatial relationships. Despite these considerations, the consistent improvement demonstrated by 2D-TPE across various tasks underscores its effectiveness in enhancing table structure perception.

## 5.3 Analysis

To gain deeper insights into the effectiveness and mechanics of our proposed 2D-TPE method, we conduct a comprehensive analysis encompassing several key aspects: investigation of its scalability regarding table sizes (§5.3.1), and the validation of its design choices (§5.3.2), its performance when based on larger models (§5.3.3), , and efficiency (§5.3.4).

### 5.3.1 Size Scaling.
To assess the robustness and scalability of 2D-TPE, we conducted size scaling experiments on HiTab. The goal was to determine how well 2D-TPE and other approaches manage tables of increasing complexity and size, crucial for real-world applications with varying table dimensions.

Specifically, we systematically expanded each original table by adding additional tables to both sides, considering three configurations: the original "0+0", one table on each side "1+1", and two tables on each side "2+2", with unchanged questions and answers. More details are in Appendix A. This method tracks performance as table width grows. As Table 4 shows, average column numbers increased from 8.1 to 21.9 and 35.8, and average table lengths (in tokens) rose from 657 to 1,647 and 2,637, respectively.

Table 4 shows **2D-TPE's superior performance and scalability over baselines**. While all baselines show a significant accuracy drop with larger tables, 2D-TPE remains relatively stable. In contrast, the row-wise traversal method, initially comparable to 2D-TPE, dropped dramatically from 57.65% to 27.52% in the "2+2" setting. Similar trends are observed with other methods. 2D-TPE's consistent performance across various table sizes highlights its versatility for diverse table understanding tasks.

Furthermore, Figure 3 illustrates **the growing advantage of 2D-TPE over three representative baselines as the number of rows or columns increases across three realistic datasets** (see Appendix A for full dataset and baseline results). For instance, on FeTaQA, *Row-wise Traversal* exhibits a mere 0.15% decrease in BLEU-4 score compared to 2D-TPE when the table comprises fewer than 4 columns. However, this gap amplifies to a substantial 2.40% as the number of columns exceeds 8. Similarly, the M-RoPE baseline on the ColType dataset initially lags behind 2D-TPE by 1.84% in the F1 score for tables with fewer than 15 rows, but this deficit exacerbates to 3.13% for tables with more than 30 rows, emphasizing the efficacy of 2D-TPE in dealing with larger tables.

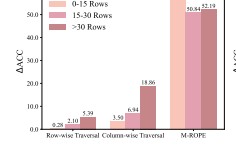 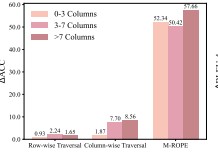 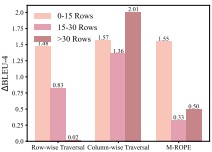 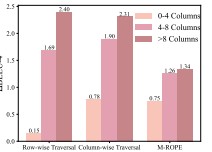 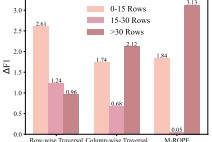 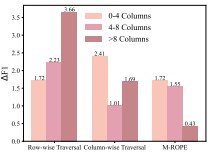

**(a) HiTab-Row**     **(b) HiTab-Column**     **(c) FeTaQA-Row**     **(d) FeTaQA-Column**     **(e) ColType-Row**     **(f) ColType-Column**

**Figure 3: Performance advantages (Δ) of 2D-TPE over three representative baselines varying with the number of rows or columns. The thresholds for stratifying tables are determined to ensure a balanced distribution of data volumes.**

**Table 5: Results (%) of the ablation study.**

| Method | HiTab ACC | EntLink ACC | FeTaQA BLEU-4 | RelExtra F1 | ColType F1 |
|---|---|---|---|---|---|
| **2D-TPE** | 68.19 | 84.10 | 65.70 | 94.18 | 86.68 |
| w/o $\mathcal{L}_{\text{ENT}}$ | 66.78 | 83.96 | 65.53 | 94.15 | 84.41 |
| w/o router | 67.79 | 83.61 | 65.34 | 93.95 | 84.49 |

**Table 6: Results (%) on Llama-3-8B-Instruct. The *italic* results are directly taken from Zhang et al. [58].**

| Method | HiTab ACC | FeTaQA BLEU-4 | RelExtra F1 |
|---|---|---|---|
| **Row-wise Traversal** | 70.75 | 66.83 | 93.25 |
| **Column-wise Traversal** | 62.52 | 66.98 | 93.44 |
| **Constrained Attention** | 16.86 | 66.17 | 91.79 |
| **TABBIE** | 62.21 | 63.17 | 92.32 |
| **M-RoPE** | 32.80 | 66.70 | 93.52 |
| **TableLlama** | *64.71* | *39.05* | *91.95* |
| **GPT-4** | *48.40* | *21.70* | *52.95* |
| **2D-TPE** | **71.39** | **67.31** | **93.81** |

*5.3.2 Ablation Study.* We verify the effectiveness of the router and the auxiliary loss $\mathcal{L}_{\text{ENT}}$ by removing them from 2D-TPE, respectively. When removing the router, we set $r^h_{m,j}$ to 1 in Eq. 7.

As observed in Table 5, removing the router leads to significant performance degradation on all tasks, indicating that interference between information from different orders hinders the model's ability to use spatial information effectively. Additionally, removing $\mathcal{L}_{\text{ENT}}$ also leads to performance drops, particularly on HiTab and ColType, indicating that the loss helps the model to distinguish spatial information from different orders explicitly. These results suggest that sharper order selections are needed to clarify the focus of each token to accurately understand table structures.

*5.3.3 Scaling to Larger Models.* To more convincingly demonstrate 2D-TPE's effectiveness, we replace the base model with Llama-3-8B-Instruct [15], supporting up to 8,096 tokens, and maintain settings from §5.1.3. Furthermore, we involve TableLlama (7B) [58] and GPT-4 as additional baselines. Due to resource limitations, we do not present the results for EntLink and ColType.

As shown in Table 6, our method still outperforms existing methods when built upon larger models, indicating the strong scalability of 2D-TPE to integrate information from multiple dimensions for perceiving and understanding table structures.

**Table 7: Efficiency investigation of 2D-TPE compared with the vanilla Transformer with Row-/Column-wise traversal. The subscripts indicate the factor by which 2D-TPE is larger than the vanilla Transformer.**

| Model | Parameter | TFLOPs | Memory (GB) | Time (Second) |
|---|---|---|---|---|
| **Vanilla** | 2.7B | 13.65 | 6.83 | 0.45 |
| **2D-TPE** | 2.7B$_{+0.05\%}$ | 13.89$_{+1.7\%}$ | 6.95$_{+1.8\%}$ | 0.51$_{+13\%}$ |

*5.3.4 Efficiency.* For efficiency evaluation, Table 7 reports the parameters, inference TFLOPs, memory usage, and average per-example inference time of 2D-TPE and the vanilla Transformer. We calculate TFLOPs using DeepSpeed FLOPs profiler [12], and memory consumption using PyTorch toolkits [40].

The results demonstrate the comparable computational efficiency of 2D-TPE with the vanilla Transformer, with almost the same number of parameters, only a negligible increase in inference TFLOPs and memory usage ($\leqslant$ 2% for both). Moreover, the average inference time of 2D-TPE is only marginally higher ($\sim$13%) than that of the vanilla Transformer. These efficiency metrics highlight the computational feasibility of incorporating 2D-TPE into existing Transformer-based architectures, without incurring significant computational overhead. Notably, the minimal additional computational cost of 2D-TPE is well justified by its substantial performance gains in capturing table structures, as shown in our extensive experiments.

## 5.4 Case Study

Appendix C presents several illustrative cases for the proposed proxy tasks and evaluation benchmarks, providing empirical insights into the efficacy, principle, and advantages of 2D-TPE.

## 6 Conclusion

In this work, we introduced 2D-TPE, a novel two-dimensional positional encoding method designed to enhance LLMs' ability to reason over tabular data. By enabling the dynamic selection of permutation orders for context perception, 2D-TPE effectively preserves the spatial relationships within table structures, addressing a critical limitation of conventional flattening approaches. Our extensive experiments across various tabular tasks demonstrate the superiority of 2D-TPE over strong baselines, underscoring the importance of maintaining structural integrity in table representation. Future work may explore additional permutation orders and extend the application of 2D-TPE to other structured data types, further enhancing the capabilities of LLMs in processing complex, multi-dimensional information.

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

## A Details for Size Scaling

To evaluate the robustness and scalability of the proposed 2D-TPE method, we devised a systematic approach to generate increasingly complex table structures by expanding the original tables. This process, illustrated in Figure 4, involves concatenating additional tables from other examples to the left and right sides of the original table, effectively increasing its width.

We considered three settings: the original table (denoted as "0+0"), inserting one table on each side ("1+1"), and inserting two tables on each side ("2+2"). This expansion strategy allows us to methodically increase the table dimensions while preserving the original questions and answers, enabling a controlled analysis of how different approaches handle tables of varying complexity.

Figure 4 elegantly depicts the construction process, with the original table at the center and the concatenated tables represented by different colors. The table set contains all tables from the HiTab training set except the original table. The "Truncated"/"Repeated" operation indicates truncating/repeating the table into the same number of columns as the original table. This systematic approach ensures a fair comparison across different table sizes, providing valuable insights into the scalability and adaptability of the proposed method in handling real-world scenarios where table dimensions can vary significantly.

It is noteworthy that the number of examples for training, validation, and testing in the size scaling experiments is less than that of the original HiTab test set. This is because we only retained examples with a sequence length not exceeding 4,096 in the "2+2" setting,

ensuring computational feasibility while preserving a diverse and challenging evaluation set.

Moreover, Table 8 provides a detailed display of 2D-TPE outperforming the complete baselines across five datasets as the number of rows or columns increases. Almost all baselines increasingly lag behind 2D-TPE with the growth in rows or columns, demonstrating the scalability and effectiveness of 2D-TPE, particularly in handling larger datasets.

## B  Hyper-parameter Sensitivity

Figure 5 illustrates the influence of the hyper-parameter $\lambda$ from Eq. (15) on our method's performance. Notably, as $\lambda$ surpasses 1, a significant performance decline is observed with increasing $\lambda$, suggesting that $\mathcal{L}_{\text{ENT}}$ should not excessively impact the model's standard training loss. Additionally, when $\mathcal{L}_{\text{ENT}}$ is too small, the performance slightly lags compared to $\lambda = 1$, indicating that $\mathcal{L}_{\text{ENT}}$ helps the model better differentiate information from two dimensions to enhance understanding of table structures. For simplicity, we fix $\lambda$ to 1 in our experiments.

## C  Case Study

We present cases for several representative tasks to illustrate the advantages of 2D-TPE in capturing tabular structures.

### C.1  Counting-Stars

The task requires LLMs to identify all cells that contain a designated star symbol within the same row or column as a specified reference cell. Table 9 presents a specific case with a 20×20 table from the test set. The question is, "What stars are in the same row and column as the number 377?" The answers provided by different methods are:

- 2D-TPE: [3★, 4★, 2★, 9★], ✓
- *Row-wise Traversal*: [1★, 1★, 3★], ✗
- *Column-wise Traversal*: [1★, 4★, 2★, 9★], ✗
- *Constrained Attention*: [3★, 9★, 2], ✗

This case study highlights the limitations of existing approaches. The *Row-wise Traversal* and *Column-wise Traversal* methods can only identify stars along their respective spatial dimensions, completely failing to capture information from the other dimension. Furthermore, the *Constrained Attention* approach struggles due to its attention pattern deviating significantly from the vanilla Transformer architecture. In contrast, our proposed 2D-TPE method accurately identifies all star symbols in the same row and column as the reference cell (377), demonstrating its robust reasoning capabilities and effective preservation of the two-dimensional table structure.

### C.2  Locating-Values

The Locating-Values task serves as a rigorous test for evaluating the multi-hop reasoning capabilities of various methods. This task requires locating the value of a cell that is a specified number of rows and columns away from a given reference cell. Table 10 presents a compelling case from the test set for the Locating-Values task. The given question is: "What is the value 3 columns to the right of and 13 rows below ★?" This query demands precise spatial reasoning and the ability to integrate information from both row

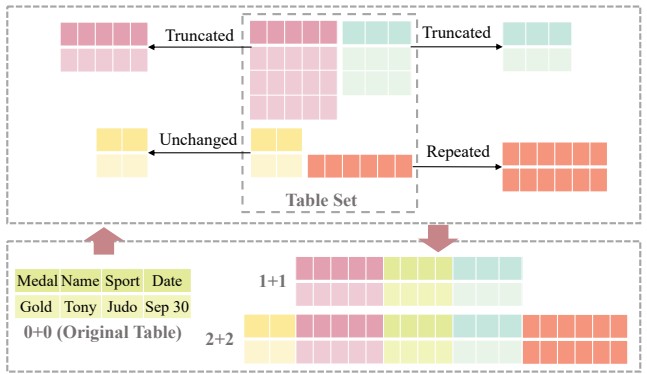

**Figure 4: Table expansion for size scaling.**

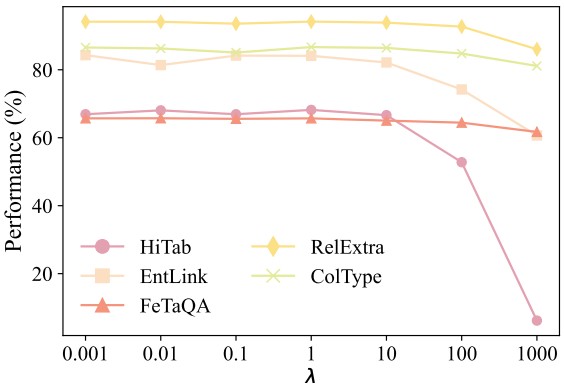

**Figure 5: Impact of the hyper-parameter $\lambda$. Specifically, we plot the change in ACC for datasets HiTab and EntLink, BLEU-4 for FeTaQA, and F1 for RelExtra and ColType as $\lambda$ varies.**

and column dimensions accurately. Different methods produced the following answers:

- 2D-TPE: 360, ✓, highlighted in yellow in Table 10
- *Row-wise Traversal*: 481, ✗, highlighted in red
- *Column-wise Traversal*: 214, ✗, highlighted in red
- *Constrained Attention*: 166, ✗, highlighted in red

This case study illustrates the effectiveness of 2D-TPE in integrating information from both row and column dimensions, enabling accurate value localization. In contrast, *Row-wise Traversal* and *Column-wise Traversal* exhibit significant limitations in handling such complex tasks due to their focus on a single dimension. Similarly, *Constrained Attention* struggles to provide correct answers in tasks requiring precise spatial reasoning.

### C.3  Case Study on Evaluation Benchmarks

We use an example from the HiTab test set to illustrate a case study on evaluation benchmarks. As shown in Table 11, HiTab is a hierarchical table dataset where solving problems requires reasoning

**Table 8: Results of 2D-TPE surpassing baselines with increasing rows and columns across various datasets. RT represents *Row-wise Traversal*, CT stands for *Column-wise Traversal*, and CA denotes *Constrained Attention*.**

| Method | HiTab ACC (%) ↑ | | | EntLink ACC (%) ↑ | | | FeTaQA BLEU-4 (%) ↑ | | | RelExtra F1 (%) ↑ | | | ColType F1 (%) ↑ | | |
| Row | 0-15 | 15-30 | >30 | 0-16 | 16-24 | >24 | 0-15 | 15-30 | >30 | 0-15 | 15-25 | >25 | 0-15 | 15-30 | >30 |
|---|---|---|---|---|---|---|---|---|---|---|---|---|---|---|---|
| RT | 0.28 | 2.10 | 5.39 | 0.48 | 0.40 | 3.05 | 1.48 | 0.83 | 0.02 | 0.64 | 0.13 | 0.93 | 2.61 | 1.24 | 0.96 |
| CT | 3.50 | 6.94 | 18.86 | 1.86 | 1.58 | 0.34 | 1.57 | 1.36 | 2.02 | 0.03 | 0.50 | 1.81 | 1.74 | 0.68 | 2.12 |
| CA | 45.10 | 44.12 | 48.82 | 2.54 | 4.74 | 0.00 | 0.97 | 0.67 | 2.56 | 11.78 | 12.05 | 16.98 | 4.79 | 3.30 | 4.84 |
| TABBIE | 4.48 | 4.63 | 11.78 | 5.75 | 8.30 | 15.76 | 2.72 | 2.06 | 3.56 | 1.86 | 1.26 | 3.27 | 3.15 | 1.12 | 2.88 |
| M-RoPE | 56.44 | 50.84 | 52.19 | 2.03 | 2.37 | -1.19 | 1.55 | 0.33 | 0.50 | 1.66 | 0.20 | 1.49 | 1.84 | 0.05 | 3.13 |

| Method | HiTab ACC (%) ↑ | | | EntLink ACC (%) ↑ | | | FeTaQA BLEU-4 (%) ↑ | | | RelExtra F1 (%) ↑ | | | ColType F1 (%) ↑ | | |
| Column | 0-3 | 3-7 | >7 | 0-4 | 4-7 | >7 | 0-4 | 4-8 | >8 | 0-4 | 4-7 | >7 | 0-4 | 4-8 | >8 |
|---|---|---|---|---|---|---|---|---|---|---|---|---|---|---|---|
| RT | 0.93 | 2.24 | 1.65 | 0.00 | 2.34 | 2.05 | 0.15 | 1.69 | 2.40 | 0.34 | 0.55 | 1.15 | 1.72 | 2.23 | 3.66 |
| CT | 1.87 | 7.70 | 8.56 | 0.42 | 0.79 | 4.62 | 0.78 | 1.90 | 2.31 | 0.17 | 0.37 | 0.20 | 2.41 | 1.01 | 1.69 |
| CA | 39.25 | 44.12 | 48.05 | 2.94 | 1.05 | 2.56 | 0.83 | 0.91 | 1.50 | 12.38 | 12.33 | 11.79 | 2.96 | 5.48 | 3.55 |
| TABBIE | 2.80 | 5.60 | 6.91 | 7.76 | 11.81 | 10.77 | 1.87 | 3.14 | 1.85 | 1.90 | 1.39 | 3.59 | 1.50 | 3.49 | 2.20 |
| M-RoPE | 52.34 | 50.42 | 57.66 | 1.47 | 0.39 | 0.51 | 0.75 | 1.26 | 1.34 | 0.98 | 1.39 | 2.11 | 1.85 | 1.55 | 0.43 |

**Table 9: A case for *Counting-Stars*.**

| | | | | | | | | | | | | | | | | | | | |
|---|---|---|---|---|---|---|---|---|---|---|---|---|---|---|---|---|---|---|---|
| 337 | 229 | 8★ | 575 | 1 | 764 | 967 | 880 | 540 | 979 | 932 | 5★ | 935 | 813 | 480 | 829 | 685 | 9★ | 522 | 365 |
| 377 | 960 | 436 | 413 | 470 | 330 | 433 | 776 | 62 | 326 | 335 | 777 | 906 | 985 | 215 | 3★ | 987 | 640 | 434 | 61 |
| 479 | 1★ | 793 | 7★ | 462 | 210 | 97 | 1★ | 908 | 675 | 912 | 493 | 304 | 671 | 416 | 983 | 458 | 515 | 954 | 614 |
| 7 | 195 | 825 | 949 | 962 | 278 | 692 | 123 | 474 | 681 | 516 | 7★ | 919 | 589 | 8★ | 178 | 282 | 530 | 783 | 5★ |
| 411 | 893 | 1★ | 41 | 2★ | 531 | 6 | 770 | 769 | 157 | 743 | 174 | 707 | 701 | 403 | 191 | 276 | 443 | 1★ | 316 |
| 796 | 127 | 901 | 865 | 528 | 974 | 502 | 313 | 518 | 71 | 565 | 684 | 486 | 34 | 752 | 400 | 803 | 4★ | 444 | 253 |
| 24 | 401 | 538 | 773 | 922 | 924 | 968 | 972 | 7★ | 2★ | 978 | 420 | 448 | 471 | 35 | 861 | 896 | 3★ | 379 | 652 |
| 150 | 38 | 843 | 527 | 818 | 50 | 226 | 963 | 943 | 676 | 6★ | 789 | 152 | 428 | 1★ | 79 | 617 | 265 | 175 | 249 |
| 4★ | 388 | 981 | 69 | 546 | 33 | 814 | 132 | 660 | 476 | 315 | 693 | 231 | 654 | 243 | 452 | 677 | 146 | 5★ | 148 |
| 2★ | 1★ | 17 | 520 | 993 | 135 | 236 | 172 | 699 | 7★ | 720 | 618 | 610 | 72 | 947 | 384 | 217 | 627 | 651 | 39 |
| 581 | 874 | 22 | 862 | 496 | 887 | 914 | 232 | 832 | 672 | 756 | 378 | 30 | 8★ | 254 | 582 | 8★ | 872 | 6★ | 32 |
| 9★ | 563 | 495 | 457 | 111 | 6★ | 8★ | 584 | 980 | 237 | 392 | 439 | 524 | 995 | 110 | 288 | 161 | 583 | 824 | 807 |
| 994 | 368 | 722 | 406 | 988 | 5★ | 279 | 534 | 257 | 833 | 702 | 782 | 989 | 831 | 8★ | 899 | 2★ | 511 | 203 | 328 |
| 103 | 742 | 842 | 630 | 8★ | 349 | 7★ | 781 | 812 | 792 | 119 | 285 | 556 | 2★ | 289 | 658 | 567 | 381 | 442 | 166 |
| 482 | 594 | 601 | 398 | 628 | 7★ | 826 | 736 | 656 | 372 | 1★ | 679 | 598 | 158 | 881 | 3★ | 645 | 29 | 117 | 418 |
| 353 | 408 | 2★ | 332 | 964 | 469 | 704 | 268 | 3★ | 312 | 389 | 688 | 4★ | 871 | 44 | 306 | 139 | 192 | 606 | 317 |
| 258 | 751 | 678 | 566 | 6★ | 984 | 228 | 625 | 248 | 6★ | 591 | 255 | 5★ | 245 | 118 | 491 | 114 | 551 | 877 | 855 |
| 206 | 790 | 194 | 5★ | 143 | 631 | 510 | 996 | 149 | 561 | 405 | 219 | 290 | 147 | 274 | 4★ | 5★ | 66 | 758 | 370 |
| 920 | 760 | 160 | 2★ | 532 | 759 | 5★ | 6★ | 354 | 63 | 725 | 52 | 931 | 969 | 23 | 16 | 196 | 42 | 422 | 915 |
| 281 | 473 | 181 | 76 | 905 | 991 | 956 | 965 | 6★ | 595 | 700 | 3★ | 990 | 4★ | 870 | 202 | 51 | 834 | 999 | 464 |

**Table 10: A case for *Locating-Values*.**

| | | | | | | | | | | | | | | | | | | | |
|---|---|---|---|---|---|---|---|---|---|---|---|---|---|---|---|---|---|---|---|
| 135 | 493 | 589 | 262 | 865 | 329 | 121 | 250 | 925 | 478 | 474 | 55 | 345 | 503 | 298 | 765 | 727 | 294 | 687 | 414 |
| 919 | 786 | 514 | 549 | 784 | 290 | 463 | 88 | 370 | 445 | 871 | 838 | 491 | 95 | 314 | 609 | 716 | 946 | 240 | 344 |
| 886 | 600 | 22 | 688 | 432 | 825 | 909 | 288 | 763 | 124 | 902 | 690 | 58 | 339 | 922 | 430 | 821 | 680 | 647 | 372 |
| 878 | 834 | 879 | 726 | 458 | 683 | 313 | 448 | 483 | 550 | 497 | 74 | 282 | 229 | 14 | 116 | 807 | 617 | 852 | 485 |
| 993 | 13 | 791 | 962 | 173 | 223 | 166 | 189 | 711 | 513 | 677 | 401 | 571 | 440 | 415 | 419 | 976 | 38 | 125 | 826 |
| 507 | 947 | 955 | 927 | 184 | 753 | 47 | 559 | 452 | 330 | 132 | 762 | 204 | 593 | 130 | 183 | 529 | 268 | 662 | 707 |
| 725 | 263 | 969 | 644 | 920 | 83 | 234 | 438 | 980 | 65 | 692 | 369 | 5 | 757 | 159 | 766 | ★ | 54 | 348 | 918 |
| 70 | 123 | 625 | 498 | 97 | 340 | 957 | 556 | 645 | 32 | 819 | 951 | 718 | 209 | 253 | 201 | 710 | 813 | 720 | 939 |
| 145 | 817 | 629 | 963 | 862 | 568 | 869 | 239 | 895 | 199 | 940 | 850 | 661 | 526 | 913 | 742 | 621 | 412 | 274 | 811 |
| 469 | 932 | 310 | 560 | 639 | 473 | 306 | 733 | 416 | 767 | 541 | 266 | 238 | 73 | 626 | 908 | 722 | 901 | 193 | 752 |
| 891 | 646 | 252 | 270 | 495 | 364 | 208 | 163 | 244 | 839 | 24 | 462 | 101 | 565 | 235 | 34 | 540 | 164 | 4 | 297 |
| 924 | 673 | 616 | 570 | 110 | 281 | 476 | 814 | 979 | 930 | 393 | 734 | 952 | 90 | 881 | 772 | 567 | 18 | 272 | 992 |
| 92 | 696 | 751 | 35 | 758 | 760 | 543 | 428 | 883 | 701 | 349 | 133 | 890 | 859 | 309 | 273 | 592 | 806 | 931 | 354 |
| 332 | 109 | 328 | 590 | 233 | 8 | 136 | 533 | 800 | 875 | 861 | 226 | 16 | 311 | 451 | 49 | 36 | 187 | 611 | 634 |
| 283 | 122 | 907 | 975 | 603 | 105 | 185 | 259 | 597 | 477 | 104 | 146 | 308 | 770 | 615 | 591 | 731 | 780 | 873 | 632 |
| 320 | 538 | 387 | 594 | 160 | 695 | 276 | 561 | 470 | 446 | 845 | 321 | 480 | 601 | 870 | 388 | 376 | 394 | 433 | 465 |
| 247 | 377 | 312 | 799 | 554 | 241 | 39 | 755 | 608 | 443 | 291 | 479 | 652 | 596 | 40 | 152 | 983 | 117 | 481 | 214 |
| 427 | 675 | 944 | 425 | 557 | 386 | 897 | 997 | 409 | 144 | 967 | 794 | 522 | 219 | 889 | 773 | 141 | 853 | 28 | 888 |
| 395 | 186 | 6 | 779 | 519 | 112 | 508 | 866 | 749 | 546 | 490 | 833 | 456 | 950 | 176 | 670 | 472 | 3 | 30 | 76 |
| 974 | 358 | 798 | 383 | 679 | 764 | 799 | 659 | 453 | 846 | 502 | 966 | 985 | 181 | 517 | 216 | 374 | 248 | 72 | 360 |

based on both row and column headers, thus necessitating the integration of information from two dimensions. In this example, 2D-TPE accurately identifies the row header "15 to 24 years-not a visible minority" and the column header "total-female," thereby correctly locating the target cell. The effectiveness of 2D-TPE can be attributed to its dynamic routing mechanism, which enables each attention head to adaptively select the most appropriate permutation order for perceiving the context. This flexible routing strategy allows the model to seamlessly integrate information from both dimensions, facilitating accurate table comprehension.

In contrast, baselines that rely on fixed traversal orders, such as *Row-wise Traversal* and *Column-wise Traversal*, suffer from localization errors due to the loss of spatial information. These methods fail to capture the hierarchical structure of the table, leading to suboptimal performance. Furthermore, *Constrained Attention* struggles because of the significant gap between the imposed attention patterns and the model's original attention mechanism, which can hinder its ability to effectively reason over tabular data.

The superior performance of 2D-TPE indicates the importance of preserving the table structure for accurate table comprehension. By dynamically routing information flow through adaptive permutation orders, our method effectively mitigates the risk of losing essential spatial information while preserving computational efficiency, thus better preserving the table structure. This novel approach represents a significant advancement in enabling large language models to reason over tabular data, paving the way for further developments in this actively explored direction.

## C.4 Router Weights

To investigate how 2D-TPE utilizes the two permutation orders, we fine-tuned MiniCPM-2B-SFT on 10,000 4×4 table data in the *Counting-Stars* task and randomly selected a sample from 2,000 test set entries. The visualization of the table and the question is shown in Figure 6(a). For the 23rd head, at layer 2, "246" allocates a larger proportion of router weights ( specifically 51.17%) to column-wise traversal, focusing on column-level information, as illustrated in the attention map shown in Figure 6(b). Here, "6" primarily attends to "1★", "9★", and "4★", the stars in the same column as "246". By layer 27, "246" shifts its focus to row-level information by allocating 52.34% of router weights to row-wise traversal. It distributes most of its attention to "9★" and "1★" in the same row, as depicted in the attention map of Figure 6(c).

This case indicates that 2D-TPE enables the model to dynamically adjust its focus between row-wise and column-wise information

**Table 11: A Case from the HiTab Test Set. The text between "[TLE]" and "TAB" is the caption for the table.**

| Table | |
|---|---|
| **Table** | [TLE] The table caption is this table displays the results of prevalence of low income. the information is grouped by low income (appearing as row headers), total, canadian-born, immigrant, female and male, calculated using percentage units of measure (appearing as column headers). [TAB] |

| low income | total | | canadian-born | | immigrant | |
|---|---|---|---|---|---|---|
| | **female** | **male** | **female** | **male** | **female** | **male** |
| | percentage | | | | | |
| **total age groups** | | | | | | |
| visible minority | 21.9 | 21.1 | 19.3 | 18.5 | 22.0 | 21.0 |
| not a visible minority | 14.3 | 12.2 | 14.2 | 12.2 | 14.3 | 12.3 |
| **under 15 years** | | | | | | |
| visible minority | 25.4 | 25.2 | 22.3 | 21.8 | 34.3 | 36.2 |
| not a visible minority | 15.2 | 15.2 | 14.9 | 14.9 | 26.1 | 25.7 |
| **15 to 24 years** | | | | | | |
| visible minority | 26.3 | 26.2 | 18.6 | 17.9 | 29.2 | 28.6 |
| not a visible minority | 15.8 | 13.7 | 15.4 | 13.3 | 20.8 | 18.7 |
| **25 to 54 years** | | | | | | |
| visible minority | 20.7 | 19.3 | 12.6 | 11.1 | 21.3 | 19.8 |
| not a visible minority | 12.7 | 11.2 | 12.5 | 10.9 | 14.3 | 13.7 |
| **55 to 64 years** | | | | | | |
| visible minority | 17.1 | 16.8 | 17.3 | 16.9 | 17.0 | 16.7 |
| not a visible minority | 14.4 | 13.2 | 14.5 | 13.2 | 13.4 | 13.1 |
| **65 years and over** | | | | | | |
| visible minority | 17.3 | 14.3 | 15.1 | 9.8 | 17.4 | 14.4 |
| not a visible minority | 16.2 | 9.5 | 17.1 | 10.0 | 12.9 | 7.4 |

| Question | what was the prevalence of low income among not a visible minority women aged 15 to 24? |
|---|---|
| **Answer** | 15.8 |
| **Answers provided by different methods** | 2D-TPE: 15.8 ✓
Row-wise Traversal: 14.3 ×
Column-wise Traversal: 14.3 ×
Constrained Attention: 12.7 × |

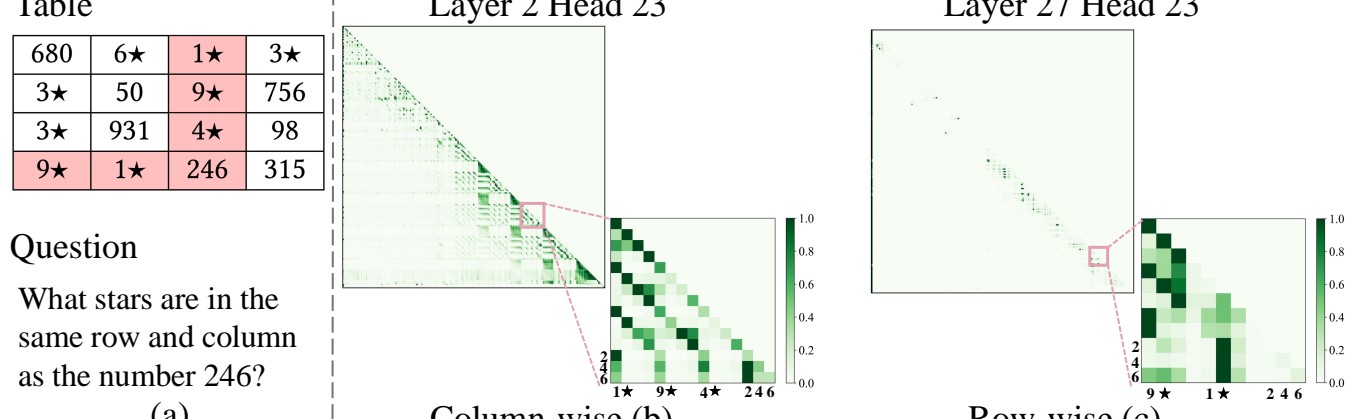

Table

| 680 | 6★ | 1★ | 3★ |
|---|---|---|---|
| 3★ | 50 | 9★ | 756 |
| 3★ | 931 | 4★ | 98 |
| 9★ | 1★ | 246 | 315 |

Question

What stars are in the same row and column as the number 246?

(a)   Layer 2 Head 23   Column-wise (b)   Layer 27 Head 23   Row-wise (c)

**Figure 6: Analysis of spatial attention distribution in the fine-tuned model for the *Counting-Stars* task.**

processing. This adaptive behavior suggests that the model learns to leverage both dimensions of the table structure effectively, depending on the specific requirements of each layer and the nature of the task at hand. In conclusion, this analysis provides valuable insights into how 2D-TPE facilitates a more comprehensive and adaptable approach to table structure perception.

Received 20 February 2007; revised 12 March 2009; accepted 5 June 2009

