# OpenReview forum: "2D-TPE: Two-Dimensional Positional Encoding Enhances Table Understanding for Large Language Models"
_ACM.org/TheWebConf/2025/Conference — WWW 2025 Poster_

### Official Review · Reviewer_WgeU · 2024-11-19

**Novelty:** 5
**Technical Quality:** 6

**Review:**

This paper identifies a common limitation in prior research:  Most existing approaches flatten 2D tables into one-dimensional sequences before inputting them into LLMs, which disrupts the spatial relationship of the tabular data and adversely affects the model's performance on table-related tasks. To address this issue, the paper proposes a novel method called **2D-TPE**, designed to enable LLMs to capture the two-dimensional structure of tables.

### **Strengths**

1. The paper is well-organized,  transitioning smoothly from foundational concepts to the proposed method, ensuring clarity throughout.
2. The proposed method 2D-TPE is insightful in its concept, while also being simple and effective in its implementation.
3. The paper provides extensive experimental validation centered on the motivations and methodology, offering strong empirical support.

### **Weaknesses**

1. **Temporal and Spatial Complexity of 2D-TPE**

    Based on the described methodology in the paper and the project implementation, introducing 2D-TPE to LLMs appears to involve repeated self-attention computations, which could double the time complexity ($O(N^2)$) compared to native LLMs. The reviewer notes the following efforts by the authors to address this issue:

    - The implementation leverages the flash attention mechanism [1] (via `self._flash_attention_forward`). However, flash attention does not reduce the time complexity of self-attention.
    - Experiments such as size scaling for tables and efficiency investigations were conducted. However, the experiments lack clarity in some specific aspects, such as : on which dataset was the *average per-example inference time* evaluated?

    The core concern here is : whether 2D-TPE introduces excessive computational overhead that compromises the LLM's ability to handle long tables or, more broadly, its performance on general long-context tasks.

2. **Generality of 2D-TPE**

    The implementation of 2D-TPE is built on TableLLaMA [2], which preprocesses tabular data from academic benchmarks to ensure uniform input formats. The reviewer wonders whether 2D-TPE can maintain its applicability under variations or perturbations in input formats, such as:

    - Tables with merged headers.
    - Tables containing empty cells.

    If the input format diverges from the assumed structure, can 2D-TPE still function effectively in broader, real-world scenarios?


[1]Dao, Tri. "Flashattention-2: Faster attention with better parallelism and work partitioning." *arXiv preprint arXiv:2307.08691* (2023).

[2]Zhang, Tianshu, et al. "Tablellama: Towards open large generalist models for tables." *arXiv preprint arXiv:2311.09206* (2023).

**Questions:**

Please refer to the example questions in Weakness above.

**Reviewer Confidence:**

3: The reviewer is confident but not certain that the evaluation is correct

**Scope:**

4: The work is relevant to the Web and to the track, and is of broad interest to the community

---

### Official Review · Reviewer_PyNC · 2024-11-23

**Novelty:** 6
**Technical Quality:** 6

**Review:**

This paper describes a novel approach of using rotary positional embeddings to embed the tokens of a table such that row wise and column wise traversals yield different positions on tokens, and attentional mechanisms attend to different traversals across heads to provide better understanding of tabular structure, as demonstrated by specific tasks such as counting stars, or locating values, or the more standard sets of tabular understanding tasks such as TableQA.  Overall, the work seems quite novel - not having much expertise in RoPE embeddings, I found the approach interesting.  Having said that though, that despite the 2DTPE name, the actual application of RoPE is still 1D, but its more in the various traversals that the 2D nature is discerned.

**Questions:**

1.  In the related work, you mentioned multiple systems such as TABERT, Stru-BERT, MATE, TURL and Tableformer.  However in the actual results, you only reported TABBIE.  Can you address why?
2.  This may reflect ignorance on my part, but table semantics is invariant with respect to column positions or row positions.  Do you consider that in the traversals you perform on the table's cell values?  Can you show that the row embeddings are in fact column invariant or that conversely your column embeddings are row invariant?
3.  I noticed that many of your 'large tables' are in fact still well within the context length of the transformer - 2.6K, and your rows and columns are in fact small (12 rows, 35 columns).  What happens when you move outside of the context length of the transformer's training?  RoPE is supposed to be less affected by that - are you as well?
4.  All your results involve fine tuning MiniCPM-2B-SFT, unless I am mistaken.  Is the inherent positional embedding in MiniCPM-2B model RoPE?  Is that why you can afford to fine tune and not have to pre-train the whole model?  What limitations exist in terms of the existing model's PE for your work to apply?
5.  Have you tested what happens when your train and test set tables vary significantly in length for the 5 tasks?

**Reviewer Confidence:**

3: The reviewer is confident but not certain that the evaluation is correct

**Scope:**

4: The work is relevant to the Web and to the track, and is of broad interest to the community

---

### Official Review · Reviewer_K4VU · 2024-12-01

**Novelty:** 5
**Technical Quality:** 5

**Review:**

This paper addresses a significant limitation in how Large Language Models (LLMs) process tabular data, specifically the loss of spatial information when converting two-dimensional (2D) tables into one-dimensional (1D) sequences. The authors propose a novel method called 2D-TPE (Two-Dimensional Table Positional Encoding), which dynamically adjusts token traversal orders (e.g., row-wise, column-wise) to preserve spatial relationships in tables. The method integrates seamlessly into existing Transformer architectures, is scalable, and offers minimal additional computational overhead.

Pros
1. The paper identifies a critical limitation in current LLM-based table understanding and provides a well-motivated solution.
2. 2D-TPE introduces a novel approach to positional encoding that dynamically adapts to the structure of tabular data.
3. The proposed method is extensively evaluated across multiple benchmarks and proxy tasks, demonstrating consistent improvements.

Cons
1. Some sections are dense and could benefit from pseudocode.
2. The paper does not provide a detailed description of the dataset, and the scalability to large datasets or those with extremely high dimensionality is not fully explored.

**Questions:**

1. Could you consider providing pseudocode for the sections that are dense to improve clarity and understanding?
2. Can you provide a more detailed description of the dataset used in the paper? Additionally, could you discuss the scalability of your approach to large datasets or those with extremely high dimensionality?

**Reviewer Confidence:**

1: The reviewer's evaluation is an educated guess

**Scope:**

3: The work is somewhat relevant to the Web and to the track, and is of narrow interest to a sub-community

---

### Official Review · Reviewer_fxkQ · 2024-12-03

**Novelty:** 4
**Technical Quality:** 3

**Review:**

This paper introduces a simple yet effective positional encoding method to address the challenge that existing methods often flatten the two-dimensional table structure into a sequence of tokens, which can severely disrupt the spatial relationships and result in an inevitable loss of vital contextual information. This paper also proposes two proxy tasks to empirically demonstrate the detrimental impact of flattening 2D table structures into 1D sequences and conducts experiments on three public datasets.
Pros:
1.	The structure of the paper is clear and relatively easy to understand.
2.	A simple yet effective positional encoding method is proposed in this paper.
3.	Extensive experiments across five benchmarks demonstrate that 2D-TPE outperforms strong baselines.
Cons:
1.	Despite Table 1 indicating that 2D-TPE outperforms three common encoding methods in the Counting-Stars and Locating-Values tasks, existing LLMs are generally capable of handling these two tasks relatively well. This raises concerns about whether the problems described in the paper actually exist.
2.	The challenges raised in the paper can be relatively easily mitigated by common techniques such as text-to-SQL, which diminishes the paper's innovativeness.
3.	The baselines compared in the experiments are too few and do not encompass various state-of-the-art table understanding methods.
4.	There is an extra comma in the first paragraph of Section 5.3.
5.	The size scaling experimental method used in the paper, namely expanding each original table by adding additional tables to both sides, is concerning. Using only this method to demonstrate that the proposed approach is suitable for complex and long tables, especially in real-word, is far from sufficient.

**Questions:**

1.	How do the best LLMs currently perform on the two additional tasks?
2.	How can the method mentioned in the paper be applied to real-world tables that are structurally complex and large in scale, and how does it perform on such long and complex tables?
3.	How do the best text-to-SQL methods currently perform on these datasets, and what are the advantages of the method proposed in the paper compared to them?
4.	How does it perform on other widely used table datasets, such as WikiTableQuestions?
5.	For complex table tasks, such as complex reasoning problems, would the proposed method in the paper provide any improvement?
6.	Would the type of tables or the type of problems affect the performance of the proposed method in the paper?

**Reviewer Confidence:**

4: The reviewer is certain that the evaluation is correct and very familiar with the relevant literature

**Scope:**

4: The work is relevant to the Web and to the track, and is of broad interest to the community

---

### Official Review · Reviewer_A7wP · 2024-12-03

**Novelty:** 4
**Technical Quality:** 4

**Review:**

This paper proposes a new attention calculation method to improve LLMs' ability in understanding 2D tables. Specifically, the tables are traversed in two directions with each permutation generating a positional embedding. Then, relative position attention (i.e., RoPE) is used to encode each permutation, followed by an aggregation of permutations with learnable weights. Overall, the paper writing is clear. The proposed method seems to perform well compared with the baselines, especially with strong scalability to larger tables.

Pros
- The usage of long-context attention in table understanding is new to me, and the experiment results show the strong performance on large table understanding. The necessity of studying large table makes sense to me and could be important for future studies.
- The two proxy tasks proposed in the paper also look interesting to me. They can demonstrate the table understanding ability while can be easily generated. They could be good sources of preliminary and case studies for future works.

Cons
- My major concern is on the technical novelty of the paper. Based on my understanding, the major idea of this paper is to apply RoPE on table understanding by attentively integrating two traversal directions of tables. The scalability of the methods also seem to come from the advantage of RoPE.
- This paper claims to be able to take a set of permutations of tabular data, but only the row-wise and column-wise traversal are studied. Given these are the most basic traversals, I feel the methodology is a little overclaiming.

**Questions:**

See cons above. Also
- In the experiment section, only a 2021 baseline is compared while there are more referenced in the related works. What is the reason other methods are not compared here?
- In line 599-600, it is mentioned that the entropy loss is to sharpen the distribution over permutations to encourage the model focusing on one permutation. The experiments also show this loss contribute to the final performance. However, intuitively I am not sure why this make sense, because understanding table needs the model to trace in both directions to locate the correct cell. Is there an explanation for this?

**Reviewer Confidence:**

3: The reviewer is confident but not certain that the evaluation is correct

**Scope:**

3: The work is somewhat relevant to the Web and to the track, and is of narrow interest to a sub-community